# Exploration of Infant Food Microbial Composition from Formal and Informal Settings Using Viable Counts and 16S rRNA Gene Amplicon Sequencing in Johannesburg, South Africa

**DOI:** 10.3390/foods12193596

**Published:** 2023-09-27

**Authors:** Wellington Torgby-Tetteh, Srinivasan Krishnamoorthy, Elna M. Buys

**Affiliations:** 1Department of Consumer and Food Sciences, University of Pretoria, Private Bag X20, Hatfield, Pretoria 0028, South Africa; wtorgbytetteh@gmail.com (W.T.-T.); srini@iifpt.edu.in (S.K.); 2National Institute of Food Technology, Entrepreneurship and Management-Thanjavur (NIFTEM-T), Thanjavur 613005, India

**Keywords:** infant food, Soweto, 16S rRNA, microbiome

## Abstract

Diarrhoea is a considerable agent of disease and loss of life in children below age five in South Africa. Soweto, South Africa is an urban township in Johannesburg, with most of its population living in informal settlements. Informal settlements in areas such as Soweto are often impoverished communities that do not get water easily, inadequate sanitation is pervasive, and poor hygiene common (risk factors for diarrhoeal diseases). Among the age groups, infants are most vulnerable to diarrhoeal infection, mainly through the ingestion of food and water. The presence of undesirable microbiota is a food safety and health challenge. This study investigated the microbiome of infant food samples collected from formal (*n* = 19) and informal (*n* = 11) households in Soweto. A non-culture-dependent technique was used to characterise the bacterial diversity and composition of the infant food samples. The results indicated that household type did not influence microbial diversity and composition in Soweto. South Africa. Firmicutes, Proteobacteria, Cyanobacteria, and Tenericutes dominated the phyla rank in food samples from formal and informal households. Potential pathogens of public health significance, including diarrhoeal disease agents such as *Salmonella* spp., *E. coli*, and *Campylobacter* spp., were detected within the foods. We concluded that the infant food samples showed rich bacterial diversity, and the presence of potential pathogens of public health significance suggests a disease risk that infants may face upon consuming the foods.

## 1. Introduction

Globally, diarrhoea remains a prime cause of sickness, disability, and death in children below age five [1,2]. The illness is common in developing countries and is closely linked to socioeconomic standing, environmental factors, and inability to access basic medical care [3]. It was projected that the disease causes 46% of mortality in Africa [1]. In South Africa, enteric infections like diarrhoea are implicated as the cause of 20% of mortalities in children below age five [4]. However, other researchers estimate diarrhoeal deaths to be between 8 and 13% [4,5,6].

Soweto, a sprawling urban settlement in Johannesburg, is South Africa’s biggest township. It has a projected 1.6 million residents, occupying 355,331 households (2019 census) [7,8]. Johnstone et al. [7] reported that Soweto has a diarrhoeal prevalence rate of 5.3%, with children more likely to present to the hospital compared to adults. The majority of Soweto’s population live in informal settlements [8], and between 600,000 and 1,000,000 residents live in abject poverty [9]. Informal settlements, such as the ones found in Soweto, are often located in areas not demarcated for residential settlements or marginal lands with nonexistent water or sanitary services. These communities are frequently associated with high population densities, unclean living conditions, inappropriate waste management, and subpar sanitation (risk factors for diarrhoea).

The aetiological agents of diarrhoea are frequently spread through the faecal–oral pathway. This is often a result of inadequate hygiene and poor sanitation, and the consumption of unclean food or drinking water coupled with inappropriate practices that make food unsafe [7]. It has been suggested that in settlements poorly supplied with water and sanitation services, households utilising resources to access improved water and sanitation avoid infection and spreading diseases like diarrhoea [10]. Since bacteria in infant food cause illness, their presence in food must be investigated.

Food consumed by infants and young children is a potential vehicle and source of infectious disease-causing bacteria. The World Health Organisation advises exclusive breastfeeding of infants for 180 days following delivery [11]. However, when infants or young children reach six months, breast milk’s energy fraction and several other nutrients are no longer sufficient to support the infant’s nutritional requirements [12]. Accordingly, there is an energy and nutritional shortfall. Infants are offered complementary foods to address the shortfall in energy and nutritional factors. The transition from breast milk consumption to solid foods exposes infants and young children to enteric infection agents such as bacteria [13]. Infant foods are nutritious, and it has been reported that they are more likely to become contaminated than those of adults [14]. Children under age 2 are much more likely to suffer from undernutrition and infection because of their rapid growth. Their susceptibility to infection is exacerbated by an immune system that is still evolving and a gut lacking in competing microflora. Infant susceptibility to infection requires that the food they ingest be free of contaminants. Aside from autochthonous bacteria associated with food, other microorganisms contaminate it along the food chain before it arrives in the household. Bacteria in food include health-promoting bacteria such as probiotics, spoilage bacteria, disease-causing bacteria, or bacteria whose current functions are unknown. Disease-causing bacteria commonly associated with food include *Salmonella* spp., *Escherichia coli*, *Aeromonas* spp. *Listeria monocytogenes*, *Campylobacter* spp., *Bacillus cereus*, *Cronobacter sakazaki*, and *Staphylococcus aureus* [15,16]. A crucial research question concerning infant food safety worth addressing is the source of contamination. Was the food contaminated with bacteria before being brought to the household? Were the bacterial contaminants introduced into the food due to household food preparation, mixing, treatment, and storage conditions? In a typical household, bacteria can be deposited on various household items including food, due to poor hygiene. Inadequate hygiene is a consequence of the unavailability of water and inadequate sanitation. These conditions favour the spread of bacteria in households. Undesirable bacteria in food compromise food safety.

The availability of water, sanitary infrastructure, and cleaning and washing practices differ in households. Likewise, food preparation plus handling practices vary across households. Limited availability of potable water, unsanitary conditions, and unhygienic habits within households spread pathogenic bacteria that cause diseases like cholera, dysentery, salmonellosis, and typhoid [10,17,18]. Households lacking access to water and sanitation are precipitated and often worsened by inadequate income, poverty, and a country’s unique sociocultural, economic, regulatory, and institutional paradigm [19]. Food processing, cooking, and mixing at home may add, reduce, or eliminate the microbiological load of the food, thereby increasing or decreasing the microbiological risks associated with the food [20]. Understanding how household types affect the microbiota of infant food before ingestion can provide a basis for advising on how to reduce contamination, contribute knowledge on food contamination, and reduce the spread of diarrhoeal diseases.

The microbiological characterisation of infant foods samples may focus on revealing the presence and role of bacteria in the causation of disease and its impact on infant health. After ingestion, the importance of bacteria derived from food as well as several environmental variables that alter the constitution and chemical pathways of the gut and its effects on our health, is becoming widely acknowledged [21].

Considered gold standards for the isolation, identification, and counting of bacteria in foods, culture-dependent methods target specific bacterial groups cultivated using nutrient-rich media and conditions designed to promote the growth of the target species [22,23]. Unfortunately, the techniques can only characterise 1% of the total number of microbes resident in samples, leaving out the more significant majority that may be viable but nonculturable (VBNC) [24,25,26]. Additionally, physiological constraints, including sublethal environmental injury and failure to compete and absorb nutrients in the growth medium, result in impaired growth [23]. This leads to an undervaluation of the number of key indicators and disease-causing bacteria found in samples and raises the need for a better technique to profile the total bacteria diversity and composition.

Bacterial detection methods that do not depend on culture, like 16S rRNA amplicon sequencing, allow us to unravel and gain insight into the total microbiome of environmental samples. Amplicon sequencing involves the targeted binding of the extremely preserved and evolutionarily enduring hypervariable sectors of the 16S rRNA genes using general primers. The amplification, sequencing, and taxonomic discrimination of the bound regions are then implemented [27]. The procedure has been widely employed to investigate the entire bacteria profile of samples [22,28,29]. Various researchers have reported Firmicutes, Proteobacteria, Tenericutes, and Cyanobacteria as bacteria dominant in foods [30,31,32,33].

The literature abounds with numerous studies examining the microbiome of foods [22,30,31,34,35]. None of these studies specifically focused on the total bacterial community of foods intended for infant consumption. Also, studies on infant foods, particularly in Africa, have focused on foodborne pathogens rather than the whole bacterial community [13]. In the quest to find the aetiological agents that cause diarrhoea in infants, the food microbiome has emerged as a candidate that deserves attention. The interplay between household type and bacterial contamination of infant foods in households in Africa is not well documented. Little or no literature is available on the total bacterial community in infant foods in Africa. Hence, there is a need to explore how household types impact bacterial diversity and composition across the food chain.

This study therefore firstly aimed to identify and compare the diversity and composition of the bacteria populations in infant foods obtained from formal and informal households in Soweto, Johannesburg, South Africa, and secondly to identify and compare the potential pathogens found in infant foods from the two household types and draw a link between the occurrence of these pathogens and disease in infants > six months of age.

## 2. Materials and Methods

### 2.1. Study Design and Enrolment of Mothers

This study was nested in a pilot project titled “Interaction between nutrition, infection, household environment, and care practices and their impact on growth and development in infants between birth and two years of age”. The Soweto Baby WASH project, as it is commonly known, aimed to document household conditions, mother and infant morbidity and illness, infant feeding, and care practices. The study evaluated the relationship between these variables and infant growth and development in the first year following birth. There were 37 time points in the Soweto Baby WASH study within a year, with participants visited at home every week from birth to 25 weeks and every two weeks from 25 weeks to 52 weeks after birth [36]. In order to recruit participants for the study, we collaborated with the Developmental Pathways for Health Research Unit (DPHRU) regarding the recruitment of mother–infant pairs to be enrolled in the microbiome study. Recruitment of the mothers was carried out at the maternity section of the Chris Hani Baragwanath Academic Hospital in Soweto. The recruitment was performed postdelivery and followed the WHO Multicentre Growth Reference Study criteria [37]. The inclusion factors for the mothers were HIV-negative, nonsmoker, without morbidity, with a term singleton pregnancy. From October 2018 to March 2019, mothers in Soweto were visited in their homes and enlisted into the study after signing informed consent forms. The researcher and one hired research assistant conducted the recruitment of the participants.

### 2.2. Collection of Infant Food Samples from Households in Soweto

Thirty food samples were aseptically collected from formal (19) and informal (11) settlements in Soweto, Johannesburg. Participants (duly scheduled for visit at the household) kept some of the food they fed to their infants and the researcher collected the food samples. The food samples were chilled (5–8 °C) and transported to the microbiology laboratory of the CFS Department, University of Pretoria. The samples were analysed immediately upon arrival at the laboratory. The types of food collected from the households are shown in Table 1.

### 2.3. Enumeration of Hygienic Quality Indicators

The standard pour plate technique was employed to estimate TVC (total viable count), EBC (Enterobacteriaceae count), and ECC (*E. coli* count). A 25 g food sample was placed into 225 mL sterilised BPW (buffered peptone water) (Oxoid, Hampshire, UK). The setup was homogenised in a stomacher for 3 min. Serial dilutions up to 10^−8^ were prepared and plated in triplicates in nutrient agar (Oxoid, Hampshire, UK). For TVC and ECC, nutrient agar (Oxoid, Hampshire, UK) and Brilliance *E. coli*/coliform agar (Oxoid, Hampshire, UK) were used and plates were incubated at 37 °C for 48 h. For EBC, violet-red bile glucose agar (Oxoid, Hampshire, UK) was used and plates were incubated at 37 °C for 24 h.

### 2.4. Extraction of DNA

A 200 mg food sample was homogenised in 100 µL sterilised ultrapure water. DNA extraction was then performed per manufacturer’s instructions. The ZymoBIOMIC^TM^DNA MiniPrep Kit manufactured by Zymo Research, Irvine, CA, USA was used.

### 2.5. Illumina High Throughput Sequencing

The amplification and sequencing of the 16S rRNA gene were undertaken at MR DNA Molecular Research (www.mrdnalab.com), Shallowater, TX, USA. To achieve this objective, the V-4 region of the 16S rRNA gene of DNA was subjected to PCR amplification using primer sets 515/806 with the forward primer barcoded. A HotStarTaq Plus Master Mix kit Qiagen (Germantown, MD, USA) was utilised as the PCR mix under the following conditions: 94 °C for 3 min, followed by 30–35 cycles of 94 °C for 30 s, 53 °C for 40 s, and 72 °C for 1 min, followed by a final elongation step of 72 °C for 5 min. To ensure that the amplification process was successfully carried out, the PCR amplicons were tested in 2% agarose gel for bands to be shown. The molecular weight and DNA concentration were used to group the samples into equimolar batches of 100. Purification of grouped samples was performed with Ampure XP beads. Illumina DNA Libraries were generated by using the gathered and purified PCR amplicons. Sequencing was conducted on an Illumina paired-end sequencing (2 × 300 bp) MiSeq platform following the manufacturer’s instructions.

### 2.6. Bioinformatics and Data Analysis

The 16S rRNA gene sequence-based taxonomy was classified using the ribosomal database. Before the quality screening, the paired-end sequences were briefly loaded into Quantitative Insights Into Microbial Ecology 2 (QIIME2) [38]. The 30 infant food DNA samples produced 4,126,806 valid sequences; following filtering and trimming, 2,591,404 high-quality reads were recovered. The DADA2 plugin was used to demultiplex and cluster the sequences [39]. Each sample’s 9854 sequences were set as the sequencing depth cutoff value. Sequences that occurred at least five times were only included during the quality filtering step, and at a 97% similarity threshold limit, the sequenced amplicons were grouped into 9854 species-level OTUs. The taxonomic composition of 16S rRNA data was determined by mapping the reads against the Greengenes Reference Data Base (version:gg-13-8-99). The Observed species OTUs, the Shannon diversity index, Faith phylogenetic diversity (Faith_PD), and Pielou’s evenness matrices were used to calculate the samples’ alpha diversity and richness index. The beta diversity between samples from the two households was assessed utilizing the Bray–Curtis index, the Jaccard distance matrix, and weighted and unweighted UniFrac distance matrices to investigate and provide probable reasons that might explain the clustering of similar bacterial communities. Principal coordinate analysis (PCoA) and boxplots were used in visualising intersample variations. The analysis of the composition of microbiomes (ANCOM) test was conducted among infant food samples to reveal variations driving differences in the relative abundance of specific taxa. The Ribosomal Database Project (RDP) classifier was used to find potential bacteria pathogen genera in the food samples using a technique previously described [28,31].

## 3. Results

### 3.1. Culture-Based Bacterial Enumeration

The TVC recovered from the formal household was 6.66 ± 1.90 log_10_ CFU/g compared to 6.91 ±1.56 log_10_ CFU/g from the informal households, with no statistical differences observed between the samples. Similarly, no differences were observed for the Enterobacteriaceae count of 4.79 ± 0.93 log_10_ CFU/g recorded in the formal household compared to 4.49 ± 1.89 log_10_ CFU/g in the informal household. In contrast, the *E. coli* count of 0.39 ± 0.93 log_10_ CFU/g recorded for the formal household was significantly different (*p* = 0.05) than the 2.03 ± 2.09 log_10_ CFU/g recovered from the informal household.

### 3.2. Microbial Diversity of the Infant Food Samples Collected from Formal and Informal Households in Soweto

#### 3.2.1. Alpha Diversity

The sequence data were rarefied to allow for the comparison of the alpha diversities of the samples (Alpha diversity estimates intrasample diversity). Rarefaction curves and boxplots were plotted for diversity metrics such as Faith_PD, Observed_OTU, and Shannon index (Figure 1). A Kruskal–Wallis pairwise comparison of the alpha diversity metrics revealed no variations (*p* > 0.05) between the formal and informal households.

#### 3.2.2. Beta Diversity

To figure out the effect that household types had on the bacteria diversity and composition in the infant foods, beta diversity metrics such as the Jaccard distance matrix, Bray–Curtis, and weighted and unweighted UniFrac indices were measured and compared for the formal and informal households. The principal coordinate analysis (PCoA) plots combined with boxplots for each beta diversity metric used to visualise differences in diversity between the formal and informal households are displayed in Figure 2. From the PCoA plots, no clusters were observed for the bacterial communities from the formal and informal households. Kruskal–Wallis pairwise comparison indicated no variation in the beta diversity between the formal and informal households.

#### 3.2.3. Analysis of Composition of Microbiomes (ANCOM)

Although no significant variations were found among the alpha and beta diversity metrices, a differential abundance test was conducted to detect differences in the key genera that are differentially abundant in infant food from the formal and informal households. The OTU table transformed using the centred log ratio (clr) method at the genera rank with zero values adjusted to one was applied. In this additive log ratio approach, genera that reject the null hypothesis > 200 times are labelled. Leuconostoc (W = 248), Campylobacter (W = 243), Micrococcus (W = 220), Bacillus (W = 206), and Bifidobacterium (W = 202) showed remarkable differences among the households (*p* = 0.05). Figure 3 shows the ANCOM volcano plot used to visualise and discriminate compositional differences between the two household types.

### 3.3. Microbiome Community Structure among Phylum and Genus Ranks in Infant Food from the Formal and Informal Households

The infant food bacterial community comprised 18 phyla, 36 classes, 30 families, 259 genera, and 406 species. Only data on the phyla and genus ranks are presented here. All 18 phyla detected were present in the infant food from the formal households, compared to 17 phyla in the infant food from the informal household. The relative abundance of the top phyla from both households are presented in Figure 4. Firmicutes, with relative abundance (46%), were the dominant group followed by Proteobacteria (37%), Cyanobacteria (9%), and Tenericutes (7%) in infant foods from formal households. Firmicutes (50%), Proteobacteria (47%), and Cyanobacteria (2%) were dominant in the infant food from the informal households.

Two hundred and fifty-nine genera were identified from clustering of the sequences of the food samples. For the formal households, 14 genera were identified, constituting about 94% of the mean relative abundance. The top genera included *Bacillus*, *Enterobacter*, *Leuconostoc*, *Vagococcus*, with 12%, 10%, 10%, and 9% relative abundance, respectively. Other top genera occurred but at subdominant levels. However, from the informal household food samples, just ten genera constituted about 94% of the relative abundances of the genus. The top genera here comprised *Weissella* (25%), *Bacillus* (17%), *Acinetobacter* (16%), *Enterobacter* (12%). Other genera occurred in subdominant levels. The relative abundance of the top genera, *Bacillus* (12%), acquired from the formal household is lower than that of the second top genera in the informal household, *Bacillus* (17%). *Weisella* (25%) was the top genera in the informal household samples, while its relative abundance was (2.7%) in the formal household. A comparison of the top ten genera between the formal and informal households is shown in Figure 4.

### 3.4. Potential Pathogen Community Structure at the Family and Genera Ranks in the Formal and Informal Households

The pathogen community structure was analysed across family, genus, and species, but only family and genera level results are presented. At the family level, 27 bacterial groups were detected and classified as potential pathogens. The top potential pathogen family within the formal household included *Enterobacteriaceae* (Proteobacteria) (47%), *Bacillaceae* (Firmicutes) (22%), *Moraxellaceae* (Proteobacteria) (16%), *Steptococcaceae* (Firmicutes) (15%), and *Pseudomonadaceae* (Proteobacteria) (2%) relative abundance among other groups. In comparison, the top potential pathogen families in the informal households were *Enterobacteriaceae* (Proteobacteria) (43%), *Bacillaceae* (Firmicutes) (25%), *Moraxellaceae* (Proteobacteria) (24%), *Pseudomonadaceae* (Proteobacteria) (9%), and *Steptococcaceae* (Firmicutes) (4%). Twenty-eight OTUs were identified and classified as potential pathogens. At the formal household level, top potential pathogen genera included *Bacillus* (24%), *Enterobacter* (20%), *Acinetobacter* (17%), *Streptococcus* (16%), and *Klebsiella* (14%) among others. Of the informal household, the top potential pathogen genera included *Bacillus* (26%), *Acinetobacter* (25%), *Enterobacter* (18%), *Klebsiella* (14%), and *Pseudomonas* (10%) relative abundance among others. A comparison of the top ten potential pathogen families and genera between the formal and informal households is shown in Figure 5.

## 4. Discussion

Elucidating the diversity and composition of infant foods is essential to understanding the possible links between microbial food safety and health. The bacterial composition of 30 food samples obtained from formal and informal households was determined as possible food safety risk factors. Potentially pathogenic bacteria were detected. The phenotypic plating method revealed high average counts of TVC and Enterobacteriaceae in the infant foods from the two household types. *E. coli* was detected in some infant food samples from both households. While high TVC and Enterobacteriaceae counts indicate unsatisfactory hygienic quality, the evidence of faecal contamination is indicated by the presence of Enterobacteriaceae or *E. coli* [40,41,42]. Culture plating can be used to benchmark metagenomic studies. For the metagenomic study, the bacterial variation within the infant food samples drawn from formal and informal households was examined through alpha diversity analysis, including microbial richness after rarefying OTU tables at 9615 sequences. Rarefaction curves reached a plateau, indicating that sample species richness and probably the original community from which the samples were drawn were sufficiently covered [43]. Alpha diversity explored using Faith_PD, Observed_OTUs, and Shannon index revealed no variation between the food samples from the formal and informal households. Similarly, we found no variations in the beta diversity indices (Bray–Curtis, Jaccard distance matrix, weighted/unweighted UniFrac distance matrices) between the food samples from the formal and informal households. The lack of variation in the alpha and beta diversity measures might be because most of the food samples collected from the households were prepared from similar staples, making microbial composition identical [31]. Similarly, non-clustering of bacterial communities according to household type was observed in the PCoA visualisation. This suggests no rare species were observed among the two household types. The same environmental conditions (including ambient temperature) might operate in the two households in the study area; hence, the bacterial composition could not have been altered considerably [31]. Also, the small sample size used for this study did not allow for significant diversity to be detected in the samples.

Regarding the 18 phyla sequenced from the 30 infant food samples, Firmicutes, Proteobacteria, Cyanobacteria, and Tenericutes contributed 98.8% of all phyla sequenced. Firmicutes (46%) were dominant in the food samples in the formal household. Other subdominant groups were Proteobacteria (37%), Cyanobacteria (9%), and Tenericutes (7%). Similarly, Firmicutes (50%) constituted the dominant phyla present in the informal household. Other subdominant phyla included Proteobacteria (47%) and Cyanobacteria (2%). Firmicutes, Proteobacteria, Cyanobacteria, and Tenericutes have been detected in milk; Firmicutes, Proteobacteria, and Cyanobacteria in MAP mussels [33]; Firmicutes and Proteobacteria in ready-to-eat Mexican food [30]; Proteobacteria, Firmicutes [31], Actinobacteria, and Bacteroidetes in pearl millet slurries [32]. In the studies listed, Firmicutes were generally the highest occurring OTUs, followed by Proteobacteria.

The persistence and abundance of the Firmicutes in food and other environments could be ascribed to their ability to form spores [44,45]. The classes *Bacilli, Clostridia*, and *Negativicutes* are known for spore formation [44,45]. These spores may resist radiation, heat, disinfectants, and desiccation, making the Firmicutes challenging to eliminate from food and food products as they can withstand unfavourable environmental conditions. This ability to sporulate may help them survive while the vegetative phases of other non-spore-forming bacteria are destroyed during processing [46]. Regarding the bacteria domain, Proteobacteria are the largest bacterial phyla and are Gram-negative [28,29,47,48]. Their presence and abundance in food and other environments are not unexpected. Enterobacteriaceae (large family within Proteobacteria) are naturally associated with food; however, they can be introduced into the food chain as part of the process contamination [49]. Some Enterobacteriaceae, notably *Cronobacter sakazakii,* are naturally associated with plant materials. They have established survival techniques that aid them in surviving for long periods in food and resist unfavourable osmotic pressure and desiccation [50]. *Enterobacter* spp., *Yersinia* spp., and *Hafnia* spp. are ubiquitous and can be found in food. *Salmonella* spp., *Proteus* spp., and *E. coli* infection result from faecal contamination and are associated with meat and poultry. However, *Pectobacterium*, *Erwinia*, and *Brennaria* spp. are commonly seen in vegetable or plant foods; *Klebsiella*, *Citrobacter*, *Pantoea*, and *Serratia* spp., may be associated with water, soil, and plants [49].

Generally, the most abundant genera in the infant foods from the formal households included *Bacillus* as the dominant genera, followed by subdominant genera such as *Enterobacter*, *Leuconostoc*, *Vagococcus*, and *Acinetobacter*. On the contrary, *Weisella* was the most abundant genera in the informal household. *Bacillus*, *Acinetobacter*, *Enterobacter*, and *Klebsiella* were the subdominant genera present. *Bacillus*, *Weisella*, *Leuconostoc*, and *Vagococcus* are members of the phylum Firmicutes. In contrast, *Acinetobacter*, *Enterobacter*, and *Klebsiella* belong to phylum Proteobacteria, which suggests the two phyla are the most dominant in infant food samples. *Bacillus* spp., widely distributed aerobic spore-forming bacteria, are associated with growing crops, and are isolated from and occur in all categories of foods [51,52,53,54,55,56]. Likewise, *Weisella,* the dominant genera in informal households, are equally ubiquitous and found in various foods like fresh vegetables, milk, meat or meat products, sourdoughs, or other foods [57,58,59].

The analyses of potential bacteria pathogen community structures revealed the presence of potentially pathogenic bacteria in food from formal and informal households. The pathogens in food could naturally be associated with the food (intrinsic) or (extrinsic) picked up along the food harvesting and processing chains. The pathogens could come from contaminated water, the environment, or animals, acting as reservoirs [60]. Thus, the utilisation of contaminated water for preparing and mixing infant food and using unclean dishes to serve the foods could also be risk factors for disease transmission.

Bacterial phyla such as *Proteobacteria* and *Firmicutes* contain bacteria families, genera, and species with significant pathogenic and potential pathogenic ability. Most of the potential pathogen sequences identified in this study belongs to Proteobacteria and Firmicutes, consistent with previous observations [28,61]. Proteobacteria, the predominant potential pathogenic phyla across the food samples from the two households, have been associated with adverse health outcomes and are described as a “diagnostic marker of dysbiosis” [62]. When such bacterial groups with pathogenic potentials bloom in high numbers in the gut, it results in an imbalance in the ideal gut microbiome and could lead to adverse health outcomes [63]. The abundance of Proteobacteria in the infant food samples should be regarded as a health risk. Potential pathogens such as *Salmonella* spp., *E. coli*, *Klebsiella* spp., *Enterobacter* spp., *Pantoea* spp., and *Acinetobacter* spp., among others, acquired from the sequenced data present a significant food safety challenge. These organisms cause intestinal infections like diarrhoea and other intestinal infections; hence, their presence in infant foods should be a concern.

This study used the culture method to prove the presence of viable bacteria in the foods. The viable bacteria are expected to be part of the 16S rRNA amplicon sequencing. The equivalency between phenotypic culture detection techniques and 16S rRNA amplicon sequencing may be biased by the presence of difficult to culture, injured, or dormant cells [64,65]. This implies that not all the sequences acquired from the infant food samples come from viable bacteria cells. Some are due to DNA molecules of dead cells or free DNA molecules in the food. This is because like all DNA-based detection techniques, 16S rRNA amplicon and shotgun metagenomics sequencing does not distinguish between signals from viable, nonviable, dead bacteria cells or free DNA in samples [64,66]. Additionally, errors in sequencing and DNA sequence alignments could result in differences in absolute counts of the OTUs in samples [65,67,68,69,70].

Pathogens in food are a risk that requires investigation and mitigation. Unsafe food is a significant global health concern. The real-time and accurate detection of foodborne disease agents is essential to averting the incidence of foodborne disease. Next-generation technologies such as 16S rRNA pyrosequencing can detect pathogens directly from food samples, thus avoiding the laborious and time-consuming steps in traditional culture techniques.

## 5. Conclusions

This study reported the microbial quality of infant food analysed using the culture plating technique and 16S rRNA amplicon sequencing. High TVC and Enterobacteriaceae counts were observed, with no variation between the households. Also, no differences were found in diversity indices employed to distinguish the metagenomic samples between the two households under study. Although no variations were observed in the microbial diversity of the food samples, analysis of composition revealed that genera such as *Leuconostoc*, *Micrococcus*, *Bifidobacteria*, *Campylobacter*, and *Bacillus* could be the main drivers of differences in diversity between the formal and informal households. The results show that *Firmicutes* and *Proteobacteria* were the predominant phyla observed across the two households. The predominant potential pathogenic bacteria at the genus rank were *Bacillus*, *Enterobacter*, and *Acinetobacter*. Although sequences of potential pathogens of public health and food safety importance were observed in low numbers in some of the food samples examined, those sequences may not be due to viable bacteria. This is because16S rRNA amplicon sequencing captures all DNA in the foods as sequence and the result may not reflect the actual numbers of viable bacteria present in food. This introduces some level of noise in the sequence output. However, the presence of those sequences is a serious infant food safety concern. The small sample size used in this study may not have allowed for differences between the two household types to be observed. We suggest control strategies be employed when mixing, preparing, and feeding infant foods to prevent contamination of infant food within the study area.

## Figures and Tables

**Figure 1 foods-12-03596-f001:**
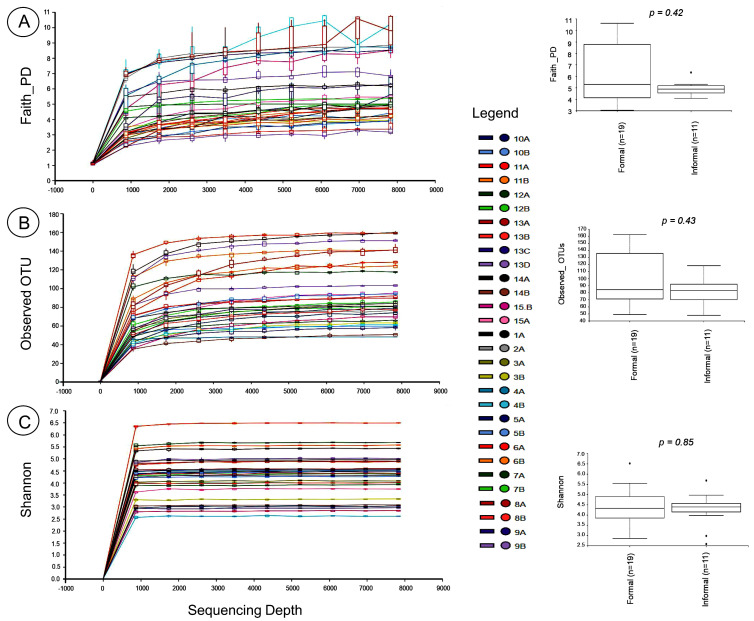
Rarefaction curves and alpha diversity boxplots for (**A**) Faith_PD, (**B**) Observed_OTU, and (**C**) Shannon diversity index. The alpha diversity indexes of the formal and informal households were compared. There was no significant difference between the two household types. The “*p*” value, which indicates statistical significance (Kruskal–Wallis pairwise comparison) between the households is shown on the boxplots (*p* < 0.05). Outliers are marked with a dot (.).

**Figure 2 foods-12-03596-f002:**
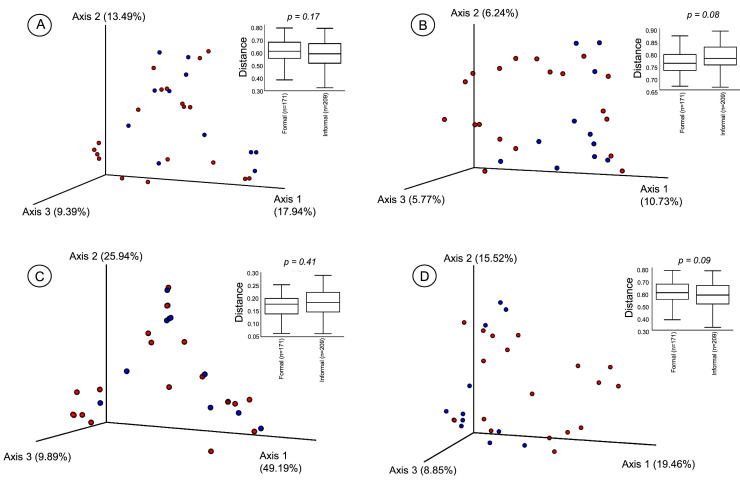
Principal coordinate analysis (PCoA) plots and boxplots for beta diversity indices based on (**A**) Bray–Curtis distance matrix (**B**) Jaccard distance matrix (**C**) weighted UniFrac distance matrix (**D**), unweighted UniFrac distance matrix. Samples from the formal households are coloured red, and informal households are coloured blue. The proportion of variation attributed to each plotted principal coordinate is shown on the axis. The beta diversity measures were compared for the formal and informal household, and no variations were observed. The “*p*” value, which indicates statistical significance (Kruskal–Wallis pairwise comparison) between the households is shown on the boxplots (*p* < 0.05).

**Figure 3 foods-12-03596-f003:**
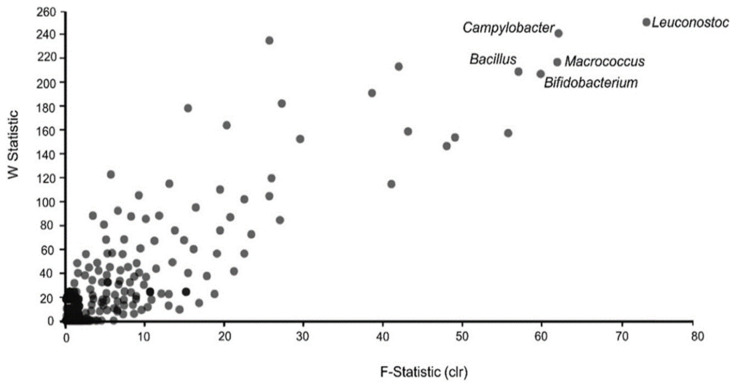
ANCOM volcano plot showing differentially abundant genera among the infant food samples. The *X*-axis indicates the centred log ratio of the transformed genus OTU table with 0 values adjusted to 1. The W statistic (see *Y*-axis) shows how many times the null hypothesis (the proportion of organisms in a group is equal to the other group) was rejected for a given species. A few genera that rejected the null hypothesis > 200 times are labelled. The dots indicate a denser bacteria (genera) population and the grey dots indicate less dense bacteria population.

**Figure 4 foods-12-03596-f004:**
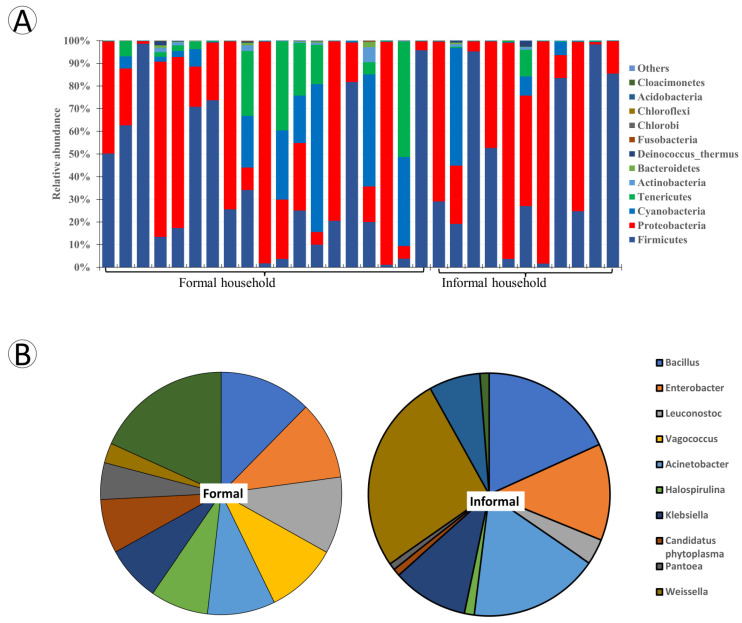
Relative abundance of the top ten bacterial groups in the infant food samples from formal and informal households: (**A**) phyla level; (**B**) genera level. All other phyla and genera are classified together as “other”.

**Figure 5 foods-12-03596-f005:**
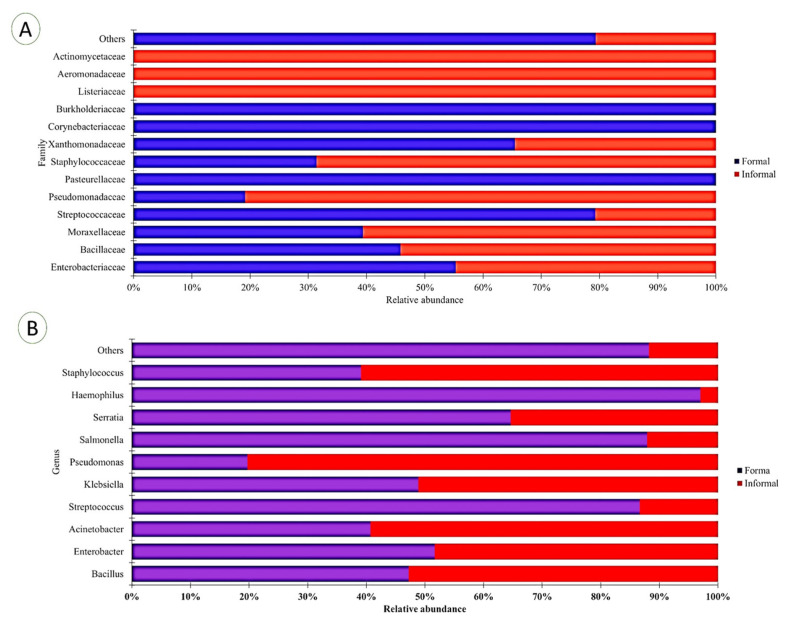
Bar charts comparing the relative abundance of the top ten potential pathogenic bacterial groups in the infant foods between the formal and informal households: (**A**) bar chart at the family level and (**B**) bar chart at the genus level. All other family and genera were categorised as “other”.

**Table 1 foods-12-03596-t001:** Types of food collected from formal (*n* = 19) and informal (11) households in Soweto.

Household	Food Type
Formal (19)	Maize porridge
	Pap/maize porridge
	Weetabix/yoghurt
	Rice and stew
	Instant maize porridge
	Pap and vegetable
	Pap and chicken
	Pap, purity
	Butternut/oats
	Sorghum porridge
Informal (11)	Pap vegetable stew
	Maize porridge
	Butternut
	Pap and beef stew
	Spaghetti in oil
	Sorghum porridge
	Pap and soup
	Sump and beans

## Data Availability

The data presented in this study are available upon request from the corresponding author.

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
