# Peer review of "Exploration of Infant Food Microbial Composition from Formal and Informal Settings Using Viable Counts and 16S rRNA Gene Amplicon Sequencing in Johannesburg, South Africa"

_foods, 2023, doi:10.3390/foods12193596_

Round 1

Reviewer 1 Report

The research article titled "16S rRNA amplicon sequencing of infant foods collected from households in formal and informal settlements in Soweto, Johannesburg" appears to be a detailed exploration of the microbiome present in infant foods in a specific region.

However, as with all articles, there are always areas for potential improvement. Below are some points of improvement:

Abstract:

  • Brevity: The abstract is quite detailed. Consider shortening it while retaining the main findings and significance of the study. An abstract should provide a snapshot of the entire study.
  • Clarity: Simplify complex sentences. For instance: "Household types impact the diversity and composition of microbiota found in foods before ingestion" can be rephrased to "Household types affect food microbiota diversity before ingestion."

Introduction:

  • Flow and Structure: The introduction could be more structured, starting with global statistics, narrowing down to South Africa, and then focusing on the local context of Soweto.
  • Relevance: Remove redundant information. For example, the repeated emphasis on diarrhea's prevalence and associated factors can be streamlined.
  • Citations: Ensure all statements that require backing have been cited appropriately.

Materials and Methods:

  • Details: While a good amount of detail is given, ensure that all methods used are described sufficiently to allow replication by other researchers.
  • Clarity in Sampling: Clarify if the food samples from formal and informal settlements were randomly selected or based on specific criteria.
  • Standardization: Mention if the same methods of collection, transportation, and analysis were used for samples from both formal and informal settlements.

Results:

  • Explicitness: In sections such as 3.2.1, 3.2.2, and 3.2.3, the term “informal households” and “formal households” is used repeatedly. It might be beneficial to define these terms early in the Results section or the Introduction, so readers don’t lose track of what is being compared.
  • Statistical Clarity: Clarify the statistical tests used to determine significance or lack thereof. For example, mention which tests were used to determine that there were "no statistical differences" in various results.

Discussion:

  • Contextual Relevance: The discussion could benefit from direct links back to the results. For example, when discussing the abundance of Firmicutes, reference specific results (e.g., the percentage abundance) to help guide the reader's understanding.
  • Depth: Delve deeper into the implications of the presence of certain bacteria, such as Enterobacteriaceae or E. coli, as markers of fecal contamination. Why is this significant, especially in the context of infant food?
  • Comparison with Prior Research: The paper frequently mentions findings from other studies. A more direct comparison, highlighting similarities and differences between the current study's results and prior findings, would strengthen the discussion.
  • Clarity: Avoid repetitive statements or those that don’t contribute new insights. For instance, the mention of Firmicutes and Proteobacteria being dominant in multiple parts of the discussion can be consolidated.

Conclusions:

  • Conciseness: Aim for a succinct wrap-up of the study's main findings. Avoid introducing new data or detailed explanations that are better suited for the Results or Discussion sections.
  • Recommendations: Expand on the suggested control strategies for preventing infant food contamination. Offer specific, actionable recommendations that can be implemented in both formal and informal settings.
  • Future Work: Consider suggesting avenues for future research, such as examining the reasons behind the observed microbial similarities in formal and informal settings or exploring the potential health implications of these microbial populations in infants.
  • Limitations: Briefly acknowledge the limitations of the study, such as the potential bias of the 16S rRNA amplicon sequencing or the sample size, to provide a balanced conclusion.

General Points:

  • Consistency: Ensure consistency in the terms used. For example, if you use "Soweto, Johannesburg," stick to that throughout instead of switching to just "Soweto."
  • Grammatical and Stylistic Errors: The document contains minor grammatical errors. Proofread thoroughly or consider professional editing.
  • Graphics and Tables: Consider including charts or graphics to present some of the quantitative data visually, especially in the results section.
  • Discussion: While it isn't presented in the provided text, ensure that the discussion section delves deep into the implications of the findings and contrasts them against existing literature.

Overall, the paper presents a detailed exploration of the microbial quality of infant foods. By addressing the aforementioned points, the authors can enhance clarity, depth, and the overall impact of their findings.

Minor editing of English language required

Reviewer 2 Report

The research manuscript submitted by Torgby-Tettch W et al. aims to analyze the bacterial communities present in infant foods collected from households in Soweto using 16S rRNA gene amplicon sequencing. The study includes 20 food samples, 19 formal and 11 informal. While the culture approach confirmed the growth of some bacteria, such as E. coli, the authors used 16S rRNA gene amplicon sequencing to characterize the bacterial composition of the samples. They found that infant food samples showed a rich bacterial diversity, and there was no significant difference between formal and informal households. However, the abundance of several bacterial genera, including Leuconostoc, Campylobacter, etc., differed. The authors also found potential pathogens in these food samples.

The manuscript is easy to follow. However, some experimental designs could be more precise and more suitable. Additionally, some results do not support key conclusions that require improvement. I have listed my comments here for the authors' consideration. 

  1. The authors need to improve their experimental designs to implicate policymaking regarding food safety and public health. It is important to note that the V4 region of 16S sequencing can only provide genus-level resolution and may not differentiate between very closely related genera, leading to inaccurate results of pathogen identification when using the RDP classifier. Therefore, the authors need to change "pathogen(s)" to "potential pathogen(s)." 
  2. The authors should provide information regarding the sampling strategy, such as choosing the food types and picking up the samples. 
  3. They should also offer more details regarding bacteria culture data in the method and results sections, especially cross-comparison between some bacterial abundance obtained from sequencing and culture. 
  4. The authors need to clarify the size of the food samples used for DNA extraction. They should also specify how many grams or milliliters of food samples were used to extract DNA and how they were homogenized. 
  5. Finally, the authors should mention which tests were used for statistical analysis for Figures 1 and 2. They should also provide the SILVA reference database version number in the method section.
  1.  
